# Advancing Web 3.0: Making Smart Contracts Smarter on Blockchain

## ABSTRACT

Blockchain and smart contracts are one of key technologies promoting Web 3.0. However, due to security considerations and consistency requirements, smart contracts currently only support simple and deterministic programs, which significantly hinders their deployment in intelligent Web 3.0 applications. To enhance smart contracts intelligence on the blockchain, we propose SMART, a plug-in smart contract framework that supports efficient AI model inference while being compatible with existing blockchains. To handle the high complexity of model inference, we propose an on-chain and off-chain joint execution model, which separates the SMART contract into two parts: the deterministic code still runs inside an on-chain virtual machine, while the complex model inference is offloaded to off-chain compute nodes. To solve the non-determinism brought by model inference, we leverage Trusted Execution Environments (TEEs) to endorse the integrity and correctness of the off-chain execution. We also design distributed attestation and secret key provisioning schemes to further enhance the system security and model privacy. We implement a SMART prototype and evaluate it on a popular Ethereum Virtual Machine (EVM)-based blockchain. Theoretical analysis and prototype evaluation show that SMART not only achieves the security goals of correctness, liveness, and model privacy, but also has approximately 5 orders of magnitude faster inference efficiency than existing on-chain solutions.

## KEYWORDS

Web 3.0, Smart contract, Blockchain, Model inference, Trusted execution environment

**ACM Reference Format:**
Anonymous Author(s). 2018. Advancing Web 3.0: Making Smart Contracts Smarter on Blockchain. In *Proceedings of Make sure to enter the correct conference title from your rights confirmation emai (Conference acronym 'XX).* ACM, New York, NY, USA, 11 pages. https://doi.org/XXXXXXX.XXXXXXX

## 1 INTRODUCTION

The concept of Web 3.0 has become very popular in recent years. It is believed that Web 3.0 can revolutionize current centralized architecture of Web systems and return the data control rights to users. Blockchain and smart contracts are one of key infrastructures promoting Web 3.0 [50]. At present, representative Web 3.0 applications are highly related to blockchains, such as Metaverse, Decentralized Finance (DeFi) [48], Non-Fungible Token (NFT), blockchain games, decentralized social media [31], and more next-generation Decentralized Applications (DApps).

One of the key issues hindering the promotion of Web 3.0 applications is the huge gap between the complex logic requirements of applications and the simple operations provided by smart contracts. Currently, powerful model-based AI algorithms drive the widespread use of many intelligent applications. If smart contracts can support AI model inference capabilities, we believe that the practicality of Web 3.0 applications will be largely strengthened and it could be a promising step to the Web 3.0 era. We can imagine that, with well-trained AI models, Decentralized Autonomous Organizations (DAOs) can filter qualified members more precisely through analyzing their transaction histories or more information; Generative AI models powered smart contracts can bring richer gameplay for blockchain games and NFTs; DeFi can explore a new unsecured lending mode by leveraging AI models to evaluate users' assets, risk, and credit before lending money. Motivated by this, we aim to make smart contracts "smarter" to build more fancy Web 3.0 applications by supporting model inference functions. Unfortunately, existing smart contracts cannot support AI model inference for two challenges: *high complexity* and *non-determinism.*

**High complexity.** The computational complexity of model inference is generally high, which is proportional to the depth and width of the neural network. If we introduce model inference functions into smart contracts, each blockchain node would bear the high computational overhead when executing and verifying smart contracts. Current blockchains have explicit complexity limits for smart contracts due to system throughput and security concerns. For example, the complexity of Ethereum smart contracts is measured through gas. Ethereum only allows up to 15-30M gas for each block [13]. As estimated in Section 6.1, even the inference computation [1] of the dedicated lightweight model, SqueezeNet, costs about 2.88G gas. It indicates that Ethereum smart contracts are far from supporting model inference. Although Harris *et al.* [18] succeeded in deploying a single-layer perceptron in the Ethereum smart contract with near 4M gas cost, they can only support very simple models in practice.

**Non-determinism.** Except for some private blockchains controlled by a single entity, all state transitions of blockchains are decided corporately through consensus mechanisms. Only if the new state is accepted by the majority of blockchain nodes, it can be "written" to the blockchain. Following this principle, smart contracts should accept deterministic code whose execution outputs are consistent in heterogeneous blockchain nodes, for achieving consensus. For example, Ethereum implements a smart contract interpreter, Ethereum Virtual Machine (EVM), and a dedicated contract programming language Solidity to ensure the output consistency of contracts. It is realized by restricting contract programs to a set

*Conference acronym 'XX, June 03–05, 2018, Woodstock, NY*
© 2018 Association for Computing Machinery.
ACM ISBN 978-1-4503-XXXX-X/18/06…$15.00
https://doi.org/XXXXXXX.XXXXXXX

---

[1] Note that model inference is making predictions based on live data and well-trained models. It should be distinguished from model training.

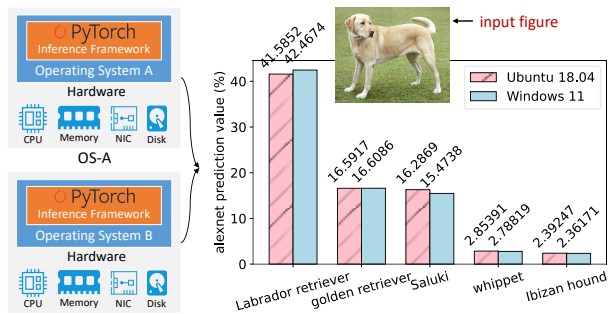

**Figure 1: Non-deterministic model inference computation.**

of deterministic opcodes (*i.e.*, integer-supported only), and executing them in the single-thread EVM. Although Hyperledger Fabric (HLF) [2] supports non-deterministic programming languages (*e.g.*, Nodejs, Java, Go) to write smart contracts (called chaincode), it demands the deterministic outputs of contracts in the block validation phase. Specifically, if the output of the smart contract is inconsistent across different blockchain nodes, then the transaction will be marked as invalid and discarded. Thus, all blockchains require deterministic outputs for achieving consensus on contract execution, which is determined by their design philosophy.

However, AI model inference is typically non-deterministic, mainly caused by several factors, such as random seeds, out-of-order parallel threads, and heterogeneous hardware [35]. Even though current two most popular inference frameworks, PyTorch and TensorFlow, both provide experimental options to make model operations deterministic, they only guarantee the outputs are reproducible on the same hardware platform and framework version [38, 45]. It is not suitable for heterogeneous blockchain nodes. As shown in Fig. 1, we also confirm the non-deterministic outputs of model inference on the same hardware platform and framework version, but different Operating Systems (OSs). Taking the AlexNet model as an example, we use the same model parameters, inputs, and PyTorch version (v1.11.0), but the inference outputs are still not the same. Thus, the inconsistent inference results cannot reach a consensus among blockchain nodes.

To enable AI capabilities in smart contracts, there are several efforts [3, 9, 18, 32] to make model inference deterministic or efficient on-chain. However, these solutions either do not support complex AI models or fail to achieve compatibility. Moreover, all these efforts compromise the accuracy of the models and cannot protect the privacy of the models.

In this paper, we propose SMART, a plug-in contract framework that supports *complex*, *non-deterministic* model inference. To achieve this goal, we design an on-chain and off-chain joint execution model to solve the above problems, while achieving good *compatibility* with existing blockchains, which is a critical feature but often overlooked. A SMART contract is composed of deterministic and non-deterministic code. We leave the deterministic code executed on-chain while outsourcing the non-deterministic code (*i.e.*, model inference) to off-chain nodes.

**On-chain.** The goal of preserving on-chain execution is to be compatible with existing blockchains and leave off-chain computing nodes *stateless*, which is different from previous off-chain

work [8, 11]. We argue that a prosperous system ecology is more difficult than its internal functions to build. Thus, one of our design goals is to be compatible with existing successful blockchains, such as Ethereum and HLF, to derive benefits from their mature ecology. We store the contract states on-chain, so the off-chain nodes can be stateless and are free to join or leave. If users do not need model inference features in their contracts, SMART contracts can naturally degrade to normal contracts by simply disabling the off-chain function. Therefore, SMART can be viewed as a pluggable module for existing blockchains.

**Off-chain.** The off-chain function handles the challenges of non-determinism and high complexity brought by model inference. To satisfy this design goal, we utilize TEE hardware to endorse the integrity and correctness [2] of the off-chain model inference. Other blockchain nodes can verify the authenticity of the inference outputs through TEE's remote attestation (discussed below) instead of re-running the model inference, thereby bypassing the non-determinism and high complexity problems simultaneously.

Besides, we utilize blockchain nodes to design a distributed attestation service to avoid relying on the centralized remote attestation server provided by TEE manufacturers, which could be the performance bottleneck and weakest link of the whole system. Considering users may want to keep their models private, we also design a TEE-aided secret key provisioning scheme, which guarantees private models are never exposed outside TEEs. To summarize, this paper mainly makes the following contributions:

- *Problem formulation.* We summarize two challenges for supporting AI capabilities in smart contracts, *i.e.*, high complexity and non-determinism, and comprehensively discuss current possible solutions and their limitations.
- SMART *framework.* We propose a pluggable contract framework to support model inference in existing blockchains. In specific, we design a TEE-based on-chain and off-chain joint execution model to enable non-deterministic, complex model inference in the contracts while compatible with existing blockchains.
- *Prototype-based evaluation.* To demonstrate the feasibility of SMART, we implement a prototype based on Intel SGX and FISCO BCOS, which is a popular open-source EVM-based blockchain platform [22] and has 2k+ stars on GitHub till now. We make the SMART prototype and experiment data open-source at https://anonymous.4open.science/r/fisco-smart-E715.

## 2 RELATED WORK AND PRELIMINARIES

### 2.1 Related Work

We discuss previous solutions and their limitations from two related work lines. One work line is committed to realizing AI capabilities in on-chain smart contracts. Another work line focuses on supporting complex smart contracts through the off-chain execution model. We present the pros and cons of on-chain and off-chain solutions in the form of radar charts in Fig. 2 and compare them with SMART.

To make model inference deterministic on-chain, the main approach is to use integers to represent floating-point numbers operations. For example, Harris *et al.* [18] use integers multiplied with $10^9$

---

[2]The correctness property is achieved under the premise of the model inference code executed inside TEE is well audited by the community.

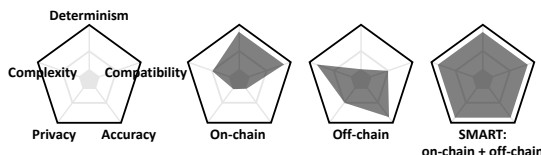

**Figure 2: Comparison of previous solutions and SMART on 5 functional indicators (Low-Medium-High).**

to represent floating-point numbers with 9 decimal places of precision. A similar solution is also adopted in [3]. Although the above on-chain solutions are well compatible with existing blockchains, due to the complexity limit of smart contracts, they cannot support complex model inference, and even decrease model accuracy. To break the complexity limit, SmartVM [32] and Cortex project [9] add convolutional instructions in EVM for supporting efficient on-chain model inference. To make the inference results consistent across different hardware, they still need to adopt quantization techniques to transform floating-point numbers into integers. These modified VMs not only cannot be compatible with EVM, but also compromise the model inference accuracy. Moreover, all these on-chain solutions are hard to prevent model privacy from disclosure.

Another type of related work focuses on supporting complex off-chain smart contracts on blockchains. Their key idea is outsourcing complex contract computation to off-chain compute nodes, such as Arbitrum [25], YODA [12], and ACE [51]. Since the number of compute nodes is generally much smaller than blockchain nodes, they have a faster convergence rate of execution and verification of complex smart contracts. Besides, Ekiden [8], FastKitten [11], and POSE [17] utilize Trusted Execution Environment (TEE)-enabled off-chain compute nodes to protect the integrity and confidentiality of contract states. However, all of these off-chain solutions do not consider non-deterministic computation, so they are not suitable to integrate model inference functions into smart contracts. Moreover, the introduction of a pure off-chain execution model generally changes the blockchain system's workflow and is not compatible with existing blockchains. Except for trusted hardware-based solutions, Zero-Knowledge Proof (ZKP) is also a potential approach to construct verifiable off-chain smart contracts. For example, both Hawk [26] and zkCNN [33] utilize ZKP techniques to realize privacy-preserving smart contracts. However, ZKP-based solutions are generally computation-consuming and need to store a large size of evaluation keys.

## 2.2 Preliminaries

TEE provides hardware-level security and privacy protection to applications. Many mainstream processors implement TEE inside to protect critical applications from attacks by untrusted hosts. For example, Intel SGX [10], one of the most widely used TEE implementations, has been embedded inside Intel Core-series and Xeon-series CPUs. Programs running inside TEE are called enclaves. Guaranteed by a series of hardware-level isolation instructions and protection policies, users can trust the output of the enclave after checking if the enclave is running inside a valid TEE.

To verify that the enclave indeed is running inside a valid TEE, TEE generally provides remote attestation based on a chain of the root. The trusted root can be TEEs' manufacturers (as Intel is to

SGX [10]) or other trusted organizations. Firstly, TEE generates an attestation quote over the enclave, which contains the enclave's output, the hash measurement of the instantiated enclave, hardware security version, denoted as $Q = (O_{\text{enclave}}, \mathcal{H}_{\text{enclave}}, \mathcal{V}_{\text{TEE}})$. The quote is signed with TEE's attestation key, denoted as $\sigma_Q^{\text{TEE}} = \text{Sig}\left(\text{sk}_{\text{attest}}^{\text{TEE}}, Q\right)$, and sent to the user over the secure communication channel. And then, the user queries the attestation collateral (e.g., TEE's key certificates) associated with the $\sigma_Q^{\text{TEE}}$ from the trusted root, and verifies that if the quote's signature is from a valid TEE. Assuming the $\sigma_Q^{\text{TEE}}$ is valid, the user can trust the enclave's outputs. We adopt Intel SGX as our TEE implementation in the prototype, but the SMART framework is TEE-agnostic. Any TEE supporting remote attestation [1, 16, 24, 28, 37] can apply to the SMART framework.

## 3 SMART OVERVIEW

Fig. 3 shows the architecture and workflow of SMART. There are three roles in the framework: clients, blockchain nodes, and TEE providers. Their functions are described as follows:

**Clients** write, deploy SMART contracts and send contract calls through blockchain nodes. If clients want to deploy SMART contracts with private models, they need to send corresponding secret shares to the Key Management Committee (KMC) for later secret key provision. **Blockchain nodes** are responsible for maintaining the security and consistency of the blockchain. In the SMART framework, blockchain nodes receive contract calls and divide them into deterministic and non-deterministic code (if exists). The deterministic code is still executed on-chain by blockchain nodes, and the non-deterministic code is outsourced to TEE providers. Besides, blockchain nodes host the distributed attestation service for TEE providers. **TEE providers** run model inference according to the contract calls signed by clients. Anyone with a TEE-enabled platform can participate in the system. TEE providers do not need to store contract states, so they can be regarded as hot-plugging modules in the SMART framework. Besides that, a quorum $t$ of TEE providers form a KMC to provision secret keys of private models, so the framework works fine as long as there are at least $t$ valid TEE providers available. We assume that communication links between clients, blockchain nodes, and TEE providers will be secured through the TLS protocol.

## 3.1 Workflow

The workflow of the SMART framework is presented in Fig. 3, we illustrate it as three parts:

**Contract creation.** Before creating a SMART contract, the client saves the model file $\mathcal{M}$ in the off-chain storage service [3] (step ①-a). If the client wants to keep the model private, he/she would encrypt the model file using a symmetric secret key $S$ before storage, denoted as $\mathcal{M}_{\text{enc}} = \text{Enc}(S, \mathcal{M})$, and sends the secret key shares to the TEE-aided KMC (step ①-b). After getting the storage URL of the model, the client specifies the model information as initial variables of the contract, such as the model name, model hash, and off-chain storage URL (step ①-c). The client deploys its SMART contract by

---

[3]The storage service can be a storage server built by the client, or a storage platform provided by a cloud vendor, as long as it can be accessed by TEE providers.

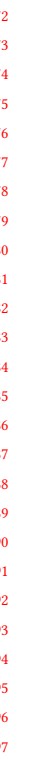

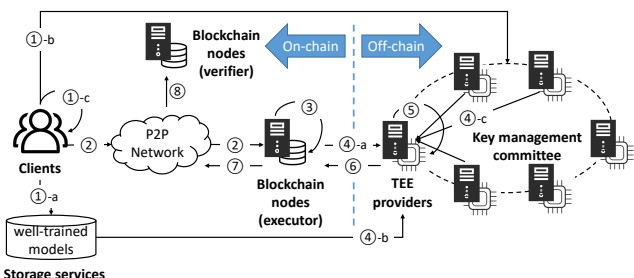

**Figure 3: Overview of SMART system model and workflow.**

posting a *contract deployment* transaction to the blockchain nodes, who will include this transaction in the blockchain (step ②).

**Contract call.** Once a SMART contract has been deployed, clients can make contract calls through RPC services at blockchain nodes (step ②). Blockchain nodes handle *contract call* requests from clients. Before executing a SMART contract call, the blockchain node first divides the contract code into the deterministic and non-deterministic parts. We use the `TEE.inference()` functions to mark the non-deterministic part in the contract (refers to Section 6.2). The former part is executed inside on-chain VMs by the blockchain node, which is the same as common contracts (step ③). The latter part is offloaded to TEE providers and executed inside secure enclaves. The blockchain node randomly chooses a TEE provider from a list of registered TEEs (refers to Section 4.2), and sends the corresponding model inference task (*i.e.*, the signed inference request) to the TEE provider (step ④-a). The TEE provider loads the model file from the off-chain storage according to the URL and checks the integrity of the obtained model file by comparing the received model hash (step ④-b). If the model is encrypted, the TEE provider needs to request the secret key of the private model from the KMC (refers to Section 4.3) and decrypts the model inside enclaves $\mathcal{M} = \text{Dec}(S, \mathcal{M}_{\text{enc}})$ (step ④-c). After that, TEE executes the inference computation inside enclaves (step ⑤). Once model inference is completed, it returns the execution results and a signed `quote` $\sigma_Q^{\text{TEE}}$ to the blockchain node (step ⑥).

When the blockchain node receives inference results from the TEE provider, it first checks the validity of $\sigma_Q^{\text{TEE}}$ through remote attestation to verify that the model inference is executed inside valid TEE hardware. And whether the hash measurement of the enclave and TEE's hardware security version match the requirement. If so, it can trust the obtained inference results. And then, the blockchain node updates the *contract call* transaction's state by aggregating the inference results and attaching the $\sigma_Q^{\text{TEE}}$ in the transaction. The nominated/leader blockchain node (executor) packs the transactions into a new block and broadcasts it to other blockchain nodes (verifiers) for verification (step ⑦).

**Block verification.** When blockchain nodes (verifiers) receive a new block, they will verify the transactions included in the block one by one. For the on-chain contract parts, the blockchain nodes verify the correctness of execution results still by simply re-executing it inside on-chain VMs. For the model inference contract parts, the blockchain nodes only verify the validity of $\sigma_Q^{\text{TEE}}$ attached in the transactions to see if the inference results are trustful (step ⑧). This bypasses the non-deterministic execution of model inference (since other nodes do not need to re-execute), and the verification

complexity is reduced from the complexity of model inference to signature verification.

## 3.2 Threat Model and Security Goals

Compared with other blockchain systems, SMART additionally introduces an attack surface of TEE providers, which could bring the following threats:

- **Inference termination and delay attacks.** Although TEE is designed to be isolated from the host OS, it also needs to interact with the host OS to access system resources such as memory, storage, and network interfaces. Thus, malicious hosts could delay forwarding or block inference requests from blockchain nodes to TEE enclaves. This attack prolongs the end-to-end latency of the transaction and even interrupts the execution process of model inference, thereby threatening the system's liveness.

- **DDoS attack against remote attestation service.** The remote attestation services are usually provided by Trusted Third-Parties (TTPs). For example, Intel provides remote Attestation Service (IAS) for SGX, whose single point of service could be the performance bottleneck of decentralized blockchain systems, and be fragile to DDoS attacks.

- **Model leakage.** Due to the transparency of blockchain and smart contracts, everyone can access model details from on-chain contracts. In practice, some clients do not want to expose their private models outside, while such a transparent architecture cannot achieve that intuitively.

Upon the aforementioned threats, we have two reasonable assumptions on TEE and blockchain: We assume there are always at least $t$ valid TEE providers available, and TEE hardware is correctly implemented and securely manufactured. We assume the blockchain will perform prescribed computation correctly and is always available. That is, there is always a majority of blockchain nodes (*e.g.*, $\geq \frac{2}{3}$) that agree on the correct outcome. Considering the above threat model and assumptions, we summarize the security goals of SMART:

- **Correctness.** The model inference will be honestly executed on given inputs and well-audited enclaves, and the integrity of inference results is guaranteed.

- **Liveness.** The valid SMART contract transactions will be eventually committed even though partial TEE providers and blockchain nodes are compromised.

- **Model privacy.** SMART offers end-to-end privacy protection for private models.

## 4 DETAILED DESIGN

### 4.1 On-chain and Off-chain Contract Execution

To make SMART framework support non-deterministic and complex model inference, while compatible with mainstream smart contract-supported blockchains, we design an on-chain and off-chain joint execution model.

TEE roles in most existing TEE-based solutions are stateful [8, 11], they need to maintain the contract states, which heavily rely on the liveness and security of TEE roles. To get rid of the need for long-lifespan TEE providers, we decouple the model inference with common smart contracts operations. Specifically, we do not

outsource the whole contract, instead just the model inference part to the TEE provider. The contract states are still stored on-chain, thus the TEE provider can be *stateless*. That is why TEE providers can join/leave the system at will without breaking the system's liveness.

We assume the SMART contract takes the form $(O, \phi_{new}) = \texttt{Contract}(I, \phi_{old})$, where $\phi$ denotes the contract state, $O$ and $I$ denote the output and input of the contract. When the blockchain node (executor) receives a *contract call* transaction, it will divide the $\texttt{Contract}$ into the on-chain part and off-chain part, denoted as $\texttt{Contract}_{vm}$ and $\texttt{Contract}_{enclave}$.

The blockchain node executes $\texttt{Contract}_{vm}$ inside on-chain VMs. For example, the on-chain contract part is executed inside EVM in Ethereum while executed inside a docker container in HLF. Thus, in the on-chain mode, the workflow is the same as existing blockchain smart contracts. And the blockchain node will outsource the $\texttt{Contract}_{enclave}$ with corresponding input data $I_{enclave}$ to a randomly chosen TEE provider for non-deterministic model inference. $I_{enclave}$ contains the client's signed *contract call* request (including model input data), the model hash, model URL, denoted as $\left(\sigma_{transaction}^{client}, \mathcal{H}_{\mathcal{M}}, \mathcal{U}_{\mathcal{M}}\right)$.

Once receiving the model inference request, the TEE provider loads the enclave program $\texttt{Contract}_{enclave}$ into TEE, and setups a TLS channel with the blockchain node. To avoid replay attacks from blockchain nodes, the TEE provider only accepts the same transaction once. After obtaining $I_{enclave}$, the enclave requests $\mathcal{M}$ from storage services according to $\mathcal{U}_{\mathcal{M}}$. The integrity of $\mathcal{M}$ would be checked inside TEE by comparing its hash measurement to $\mathcal{H}_{\mathcal{M}}$. After preparing essential input data $\sigma_{transaction}^{client}$, the enclave starts model inference with the fetched model and outputs the inference results $O_{enclave}$. To prove that the model inference is executed inside valid TEE hardware and the integrity of inference results, the enclave will provide an quote on its identity and inference outputs, $Q = \left(O_{enclave}, \mathcal{H}_{enclave}, \mathcal{V}_{TEE}, \sigma_{transaction}^{client}\right)$. Including $\sigma_{transaction}^{client}$ in the quote is to bundle the inference computation with one specific *contract call* transaction, which can avoid free-riding and false-reporting attacks from TEE providers. Then, the TEE provider returns the signed quote $\sigma_Q^{TEE}$ to the blockchain node.

The blockchain node will verify the validity of $\sigma_Q^{TEE}$ through remote attestation, check if $\mathcal{H}_{enclave}$ and $\mathcal{V}_{TEE}$ match the requirement, and $\sigma_{transaction}^{client}$ is the target *contract call* request. If all these checks pass, the blockchain node accepts the inference output $O_{enclave}$. If $O_{enclave}$ is not the final output of the contract, the blockchain node will take $O_{enclave}$ and $O_{vm}$ (if exists) as intermediate results to continue computing the final output according to the contract logic, update the contract state $\phi$ and seal them as a newly completed transaction. Note that the blockchain node could outsource multiple model inference tasks to TEE providers simultaneously, and wait for responses in an asynchronous mode, thus the off-chain inference does not congest block generation.

However, malicious host OSs could interrupt or delay TEE's model inference execution on purpose. Although they cannot forge the execution outputs or steal secrets inside TEE, such attacks could prolong the transaction latency and even break the system's liveness. To prevent endless waiting, we set a simple *timeout* policy

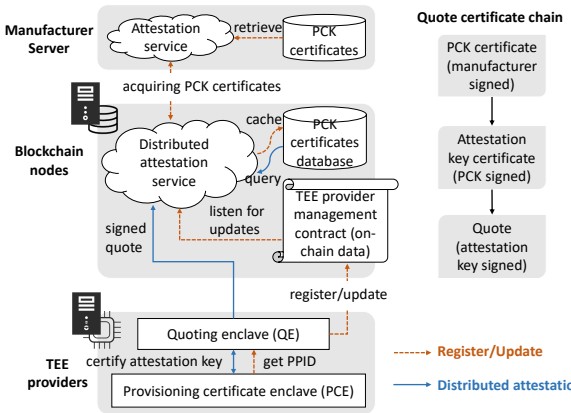

**Figure 4: Blockchain nodes assisted distributed attestation service.**

for blockchain nodes. If the waiting time of the blockchain nodes exceeds the threshold $\delta$, they will assign another TEE provider to perform the inference task and raise an evidence challenge of the abnormal TEE provider, similar to POSE [17]. If the evidence is confirmed, the abnormal TEE provider would be added to a blacklist and not used again. In this way, as long as there is at least one normal working TEE provider exists, the system's liveness is guaranteed.

## 4.2 Distributed Attestation Service

To mitigate potential DDoS attacks against centralized remote attestation services hosted by TTPs, we leverage blockchain nodes to construct a distributed attestation service that avoids the single point of failure.

The intuitive idea is that we can make blockchain nodes cache the quote certificate chain and provide quote verification services locally. The whole blockchain network could be seen as a decentralized "third party" to provide attestation services and does not need to access TEE manufacturer-provided attestation services frequently. The design of the distributed attestation service can be described as Fig. 4. We mainly group it into two flows: the register/update of TEE providers (orange lines) and the distributed attestation (blue lines).

**Register/update of TEE providers.** We require TEE providers to register their identities before participating in the system, which is to identify the validity of TEE providers and for further incentive distribution (we discuss incentives for TEE providers in Appendix A). We design the TEE Provider Management (TPM) smart contract (the implementation code refers to Appendix B) to record the Platform Provisioning ID (PPID) of TEE providers. PPID is unique to the platform and Provisioning Certificate Enclave (PCE) identity, and it is constant during the platform's lifetime. Thus, we use PPID as the unique identifier of TEE providers. The TPM contract contains two functions: $\texttt{register}$ and $\texttt{update}$.

For registering a TEE provider, the Quoting Enclave (QE) first retrieves the TEE platform's PPID from the PCE, and then the TEE provider sends its PPID to the TPM contract via invoking the $\texttt{register}$ function. Blockchain nodes keep listening to the TPM contract for updated events. If there comes a new registered TEE

provider, blockchain nodes will acquire corresponding attestation collateral from the TEE manufacturer-provided attestation service by querying with the PPID of the TEE provider. The attestation collateral contains Provisioning Certification Key (PCK) certificates, PCK certificate revocation lists, Trusted Computing Base (TCB) version, and the QE identity for platforms with TEE enabled. Blockchain nodes cache the obtained attestation collateral in their local database for later quote verification. If the TEE provider needs to update its PCK due to CPU microcode updates or PCE updates, the QE will retrieve a new PPID from the PCE and update its PPID in the TPM contract by calling the update function. Once blockchain nodes detect PPID updates, they will acquire new attestation collateral for all updated TEE platforms. It is similar to the TEE provider registration process.

**Distributed attestation.** After TEE providers register their identities on-chain, they can perform remote attestation to blockchain nodes without connectivity to manufacturer servers. In specific, the QE first generates an attestation key $\mathsf{sk}_{\mathsf{attest}}^{\mathsf{TEE}}$ (used to sign quotes) and requests the PCE to certify the attestation key (only on first deployment). Then, the QE can obtain an attestation key certificate signed by the PCE. With the attestation key certificate, the QE can generate a quote structure containing TCB version (*i.e.*, hardware security version $\mathcal{V}_{\mathsf{TEE}}$), enclave identity $\mathcal{H}_{\mathsf{enclave}}$, inference outputs $O_{\mathsf{enclave}}$, attestation key certificate, PCK, and sign the quote using the certified attestation key. Then, the TEE provider sends the signed quote to the blockchain node for remote attestation verification. When the blockchain node receives a quote, it will query the PCK certificate from its local database according to the PCK embedded in the quote, and verify the validity of the quote according to the quote certificate chain (shown as Fig. 4). Assuming the quote is certified correctly, the blockchain node will further check if $\mathcal{V}_{\mathsf{TEE}}$ satisfies the minimum value and $\mathcal{H}_{\mathsf{enclave}}$ is the desired one. Only after all these checks pass, the blockchain node can trust the inference results in the quote. To this end, the remote attestation and verification between blockchain nodes and TEE providers are completed without additional TTPs. Even if some of the blockchain nodes are temporarily unavailable due to DDoS attacks or other reasons, the distributed attestation service still works normally in the rest of the blockchain nodes. Note that we need to acquire attestation collateral from the manufacturer server when the TEE provider registers or updates their identities, so the trust root still exists. Even so, the call frequency of register/update of TEE providers is much smaller than remote attestation, thus the distributed attestation service can effectively improve the scalability and security of the system.

### 4.3 Secret Key Provisioning for Private Model

In practice, some clients may use private models or commercial models for inference and want to protect their models' privacy during outsourcing computation. A straightforward solution is that clients can encrypt their private models and only decrypt models after being loaded into TEE. However, this raises a problem: *how can the secret key to decrypt the private model be securely provided to the TEE and ensure that the key is always accessible?* To solve this problem, we design a TEE-aided Shamir's Secret Sharing (SSS) scheme to manage the secret keys of private models.

**Secret shares distribution.** Assume that SSS [40] is an ideal and perfect $(t, n)$-threshold scheme based on polynomial interpolation over a finite field $GF(q)$. We leverage $n$ TEE providers to constitute a secret key management committee. We denote the secret key of a private model as $S$. The client firstly randomly chooses $t - 1$ elements $a_1, \cdots, a_{t-1}$ from $GF(q)$ and construct the polynomial $f(x) = S + a_1 x + a_2 x^2 + a_3 x^3 + \cdots + a_{t-1} x^{t-1}$.

Then, the client computes any $n$ points out on the polynomial $f(x)$, for instance, set $x = 1, \cdots, n$ to find points $(x, f(x))$. Note that these $n$ points should be non-zero. These $n$ points are treated as $n$ secret shares of $S$ and separately transmitted to the $n$ TEEs in the committee over the secure channel.

**Key recovery.** If a TEE provider wants to make an inference on a private model, it should recover the secret key from the committee. The TEE provider firstly retrieves $t$ secret shares (*i.e.*, points) from any $t$ TEEs from the committee over the secure channel. Obtained $t$ secret shares, the secret key $S$ can be recovered using interpolation $S = f(0) = \sum_{j=0}^{t-1} y_j \prod_{\substack{m=0 \\ m \neq j}}^{t-1} \frac{x_m}{x_m - x_j}$.

Thus, the TEE provider can decrypt the private model using the recovered $S$ inside TEE, and make model inference. The authenticity of TEEs is guaranteed by the proposed distributed attestation service, so our TEE-aided SSS scheme does not need to consider malicious nodes. As long as there are at least $t$ TEEs available in the committee, the secret key can be recovered.

## 5 SECURITY ANALYSIS

Based on the threat model proposed in Section 3.2, we analyze the system security formally to show how the aforementioned security goals, *i.e.*, liveness, correctness, and model privacy, are achieved.

In the SMART framework, both blockchain nodes and TEE providers contribute to the system liveness. Previous work has proven that valid transactions will be included in the blockchain within sufficient time [21, 52], so we mainly demonstrate that TEE providers will not break the liveness of blockchain systems.

THEOREM 5.1 (SYSTEM LIVENESS). *If an honest blockchain node outsources a model inference task to TEE providers,* SMART *would complete the task within $n - t + 1$ rounds.*

PROOF. Under the assumption that network is partial synchronous, one honest blockchain node broadcasts a inference task, the other honest TEE providers can receive the task within a certain time. However, a malicious TEE provider might not response or prolong the waiting time to delay the task. Once the TEE providers accept the task, based on the *timeout* epoch $\delta$ we set up, the blockchain nodes can prevent deadlocks caused by malicious behaviors or accidents of TEE providers. The blockchain node can abandon current round and choose another TEE provider if it receives nothing from current TEE provider after waiting for $\delta$. Under the assumption that there are always at least $t$ of $n$ TEE providers working normally, the blockchain node always can find available honest TEE providers to execute model inference task. We denote $X$ as the event that the blockchain node chooses an honest TEE provider in the $k$-th round, and then we have

$$Pr(X) = \begin{cases} \prod_{k=1}^{n-t} \left(1 - \frac{t}{n-k+1}\right)^{k-1} \frac{t}{n-k+1}, & \text{if } 1 \leq k \leq n - t, \\ 1, & \text{if } k > n - t. \end{cases}$$

Therefore, SMART will complete the inference task within $n - t + 1$ rounds no matter what Byzantine behaviors happen, and the system liveness is guaranteed. □

THEOREM 5.2 (EXECUTION CORRECTNESS). *If the output of a model inference task is included in the blockchain of the majority of blockchain nodes, the execution result of model inference is correct.*

PROOF. After completing model inference, the TEE provider generates a quote $Q = \left(O_{\text{enclave}}, \mathcal{H}_{\text{enclave}}, \mathcal{V}_{\text{TEE}}, \sigma_{\text{transaction}}^{\text{client}}\right)$ to provide the inference result. Since $Q$ is signed by the TEE's attestation key $\sigma_Q^{\text{TEE}} = \text{Sig}\left(\text{sk}_{\text{attest}}^{\text{TEE}}, Q\right)$, the blockchain node can verify the validity of $Q$ through the distributed attestation. And then, the blockchain node checks if the received hash measurement of the enclave $\mathcal{H}_{\text{enclave}}$ equals the targeted one. If so, it indicates that the TEE provider executes model inference inside the correct enclave program $\text{Contract}_{\text{enclave}}$. Besides, the blockchain node checks if the version of TEE $\mathcal{V}_{\text{TEE}}$ is up-to-date and if the input data of the enclave is from the correct client and transaction $\sigma_{\text{transaction}}^{\text{client}}$. With the correct input data, TEE enclave measurement, and quote signature, SMART guarantees the *correctness* of inference results. Only if all the above checks pass, honest blockchain nodes will include $Q$ and $\sigma_Q^{\text{TEE}}$ in the new block. And previous work has proven that *if a transaction is included in a block of the blockchain of an honest node, this transaction will be ultimately persisted in every honest node's blockchain with high probability* [21]. Under the assumption of a majority of blockchain nodes are honest, the theorem is proven. □

THEOREM 5.3 (MODEL PRIVACY). *If a rational client submits a private model inference transaction, the private model parameters would not be leaked throughout the whole inference process.*

PROOF. A client will encrypt its private model using a randomly generated symmetric keys $S$ and store the encrypted model in the storage services. And then, the client generates $n$ secret shares based on $S$ and sends them to $n$ TEE providers in the KMC for secret key provision. Since the transmission of the secret shares is secured by the TLS communication (the authenticity of the TLS channels is ensured by identity attestation $\sigma_Q^{\text{TEE}}$ between TEEs and clients), the secret shares would not be disclosure. When the TEE provider needs to execute a private model inference task, it will request secret shares from randomly chosen $t$ TEE providers in the KMC to recover the secret key $S$ as described in Section 4.3. Since we assume TEE hardware is correctly implemented and securely manufactured, and $S$ is recovered inside authenticated TEE hardware, no one can know the content of $S$ including TEE providers themselves. Moreover, a rational client always does not proactively leak the secret key $S$. Therefore, as long as the encryption algorithm is secure, the privacy model cannot be compromised. □

# 6 IMPLEMENTATION AND EVALUATION

We adopt FISCO BCOS [22] as the underlying blockchain system, which is a popular consortium blockchain embedded with the EVM engine, so it is completely compatible with Ethereum smart contracts. We realize the on-chain and off-chain context switch in the Ethereum smart contract through the precompiled contract technique [15], which supports the on-chain contract calling customized off-chain programs. Thus, the off-chain programs (*i.e.*, precompiled

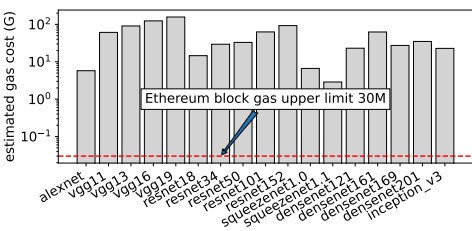

**Figure 5: Estimated gas cost for on-chain model inference.**

contracts) act as a bridge to connect blockchain nodes and TEE providers. Through it, the SMART framework can provide backward compatibility for mainstream blockchains.

We take Intel SGX as the experiment TEE hardware and use PyTorch as the model inference framework. For simplifying SGX application deployment, we leverage Graphene-SGX [47] (now called Gramine), a LibOS for TEE hardware, to support model inference inside enclaves without modifying Python code. We leverage Intel DCAP (allowing third-parties nodes to cache attestation collateral) [39] to implement distributed attestation service to certify the quote with a certificate chain rooted to an Intel-issued certificate.

We set up a four-node blockchain network and one TEE provider instance to evaluate the SMART framework. We adopt a (3, 4)-threshold SSS scheme to make secret shares and the AES-128-GCM symmetric encryption algorithm to encrypt/decrypt private models. The prototype is deployed on a laptop with an Intel i7-8750H CPU (supporting SGXv1 with FLC) and 16 GB RAM running the Ubuntu 20.04 operating system. We compare the SMART framework with existing solutions [3, 18, 32] to fully reveal the necessity and superiority of our framework.

## 6.1 Estimated Gas Cost for On-chain Inference

In order to prove the impracticality of on-chain solutions [3, 18], we estimate how much gas would be consumed to implement model inference on-chain.

Since the traditional smart contracts cannot support floating point operations, we need to use integers to represent floating point numbers during on-chain model inference [18]. We can calculate the number of multiply-accumulate operations (MACs) to approximate the complexity of AI models [44]. Taking Ethereum as an example, we can know the total cost of a ADD and a MUL operation of the EVM is 8 gas [14]. Thus, 8×#MACs is an approximate gas cost for on-chain model inference. We estimate the gas cost of dozens of popular AI models and mark the block gas limit using a red dotted line in Fig. 5. We can observe that the gas cost of all these models far exceeds the block gas limit of 30M. Even for the lightweight architecture model, SqueezeNet [23], it still consumes about 2.88G gas. These results demonstrate that on-chain model inference cannot support these mainstream models even using integers for data representation. Moreover, developers have to spend a lot of time porting models to traditional smart contracts (*e.g.*, Solidity). Therefore, the on-chain model inference schemes [3, 9, 18] are not practical. In comparison, our SMART framework bypasses the gas limit by offloading the model inference to TEE providers while retaining the authenticity of the computation. And TEE providers can use the off-the-shelf model files and inference frameworks to do the model inference computation, which largely reduces the

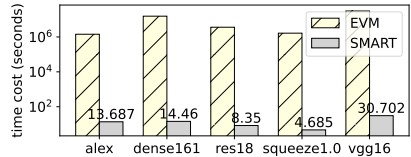

**Figure 6: Model inference time comparison of EVM-based on-chain solutions [3, 18] and SMART.**

development cost. The experimental result shows the necessity of offloading model inference to off-chain nodes.

## 6.2 Performance of SMART Contract Inference

**Model inference.** To further demonstrate the efficiency of model inference on SMART, we choose five popular AI models, *i.e.*, AlexNet [27], SqueezeNet [23], ResNet [19], DenseNet [20], and VGG [43] as the inference models, and compare the inference time of EVM-based on-chain solutions [3, 18] with SMART, as shown in Fig. 6.

We can observe that SMART's inference efficiency is about 5 orders of magnitude faster than EVM-based on-chain solutions. That is why existing on-chain solutions only can support simple models like single-layer perceptron or naive bayes classifier [18]. In comparison, SMART can complete model inference within a few seconds, which largely enhance the feasibility of model inference on blockchains.

**End-to-end latency.** To reflect the impact of introducing SMART framework in existing blockchains, we write a `SmartCall` contract that has an `inference` function purely calling the off-chain model inference function `TEE.inference()` to test the end-to-end latency of a SMART contract call (refers to Appendix C).

We evaluate the end-to-end latency of the SMART contract call under the public and private models, and compare it with SmartVM [32], an AI operators embedded smart contract VM, as shown in Fig. 7. Taking AlexNet and ResNet18 as examples, we can observe that the end-to-end latency of SMART is better than SmartVM, especially for AlexNet, there is about 2.7x improvement. Note that SmartVM does not support private model inference. Even for private model inference, SMART still has lower latency compared to SmartVM.

**Micro-performance.** We also analyze the micro-performance of SMART to show the time overhead of each component in Fig. 8.

The latency difference of contract calls between private models and public ones is caused by additional secret key provisioning and decryption operations. In our experiment setting, the TEE provider who needs to decrypt private models will set up TLS connections with 3 TEE providers in the KMC to acquire secret shares and recover the secret key $S$. We observe that the process of secret key provisioning is completed in about 5.15 seconds on average (denoted as SKP in Fig. 8). It is an acceptable time overhead.

The time overhead of quote generation (QG), verification (QV), and context switching between on-chain and off-chain (OC and CR) is negligible. In the QG and QV phases, the registered TEE provider generates a signed `quote` and sends it to the blockchain node for verification. Since the blockchain node only needs to query it from the local PCK certificate cache database for verifying the $\sigma_Q^{\text{TEE}}$ (an ECDSA-based verification), the attestation process can be completed quickly in about 0.172 seconds on average.

We can observe that the overall latency is around 75 seconds for the chosen models, of which the enclave initialization is the

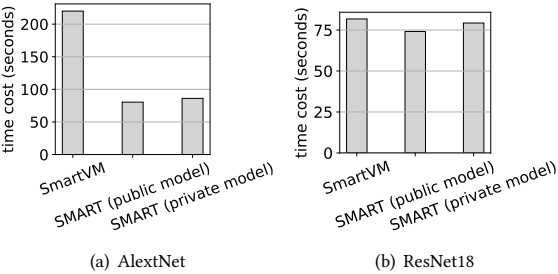

(a) AlextNet          (b) ResNet18

**Figure 7: End-to-end latency of SmartVM [32] and SMART.**

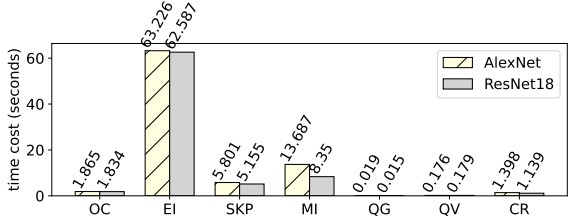

**Figure 8: Micro-performance analysis of SMART (OC=On-chain Contract Call; EI=Enclave Initialization; SKP=Secret Key Provisioning; MI=Model Inference; QG=Quote Generation; QV=Quote Verification; CR=Contract Context Recovery).**

major factor of the long latency, about 61.5 seconds (denoted as EI in Fig. 8). The reason is that we adopt Graphene-SGX [47] to bring convenience for developing SGX applications, but it takes a certain time to load massive LibOS files and check their integrity. On one hand, the latency brought by EI can be avoided through initializing the enclaves in advance. On the other hand, this overhead could be optimized by pruning unnecessary LibOS files or developing model inference enclaves from scratch using the official SDK. Besides, we can adopt some proper read/write strategies to reduce page swap cost [29]. We further discuss the potential strategies for improving TEE performance in Appendix D.

Nevertheless, the latency of off-chain model inference does not congest the blockchain throughput due to its asynchronous request-response mode. And model inference tasks can be executed in parallel (*i.e.*, multiple inference tasks can be assigned to multiple TEE nodes simutaneously), which could lift up the inference throughput from the system perspective.

## 7 CONCLUSION

In this paper, we propose the SMART framework to support model inference atop existing blockchains, which incorporates an on-chain and off-chain execution model to handle the non-deterministic and complex computation while achieving good compatibility. We also leverage blockchain nodes to host a distributed attestation service to avoid the single point of failure brought by the centralized attestation server. Besides, we design the TEE-assisted Shamir's secret sharing scheme to provide end-to-end privacy preservation for private models. Both analysis and evaluation show the security and feasibility of the proposed framework. We believe it is a worthy attempt to bring AI capabilities to existing smart contracts, which could endow smart contracts with more application space in the upcoming Web 3.0 era.

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

## A INCENTIVES FOR TEE PROVIDERS

In the SMART framework, we introduce new roles, TEE providers, to execute off-chain model inference, so we have to re-design the incentive scheme to reward TEE providers. Considering that TEE providers will register their TEE hardware identities (*i.e.*, PPIDs) before participating in the system, they would generate new blockchain accounts (or use existing blockchain accounts) and bundle them with PPIDs in the TPM contract (see Appendix B). The registered blockchain accounts are used to distribute rewards/transaction fees to TEE providers. In other words, the rewards obtained by TEE hardware execution will be sent to the TEE provider, so a rational TEE provider has less motivation to do evil.

**Table 1: The number of parameters and MACs of the models.**

|  | AlexNet | DenseNet161 | ResNet18 | SqueezeNet1.0 | VGG16 |
|---|---|---|---|---|---|
| # of Params (M) | 61.1 | 28.68 | 11.69 | 1.25 | 138.36 |
| # of MACs (G) | 0.72 | 7.82 | 1.82 | 0.83 | 15.5 |

In public blockchains like Ethereum, the blockchain nodes will be paid transaction fees for executing and packing transactions into a new block. In the same way, TEE providers should also be paid rewards for executing the off-chain part of a *contract call* transaction, *i.e.*, model inference. Similar to the gas concept in Ethereum, we quantify computation costs for model inference by the complexity of models. We adopt the number of multiply-accumulate operations (MACs) and the number of parameters of models to estimate how expensive a model inference can be. We list the number of parameters and MACs of five popular models in Tab. 1. One MAC contains an addition and a multiplication operation, *i.e.*, $a \leftarrow a + (b \times c)$. The number of MACs can roughly represent the computational complexity of the model. The number of parameters represents the space complexity of the model. Combining the computing and space complexity metrics, we can easily establish a linear mapping from model complexity to gas settings. Note that we do not give a specific mapping relation between them in this paper since it is a configurable setting. When the new block is generated, the rewards of model inference will be transferred to TEE providers automatically as transaction fees. Thus, in public blockchain systems, the TEE providers can be paid proper transaction fees according to the gas consumed during model inference.

As for the blockchain nodes and TEE providers of consortium blockchains (*e.g.*, HLF [2], FISCO BCOS [22]), they are generally hosted by several authorized organizations, so consortium blockchains do not need transaction fees to incentive participants.

## B TEE PROVIDER MANAGER (TPM) CONTRACT

The TPM contract manages the identities of registered TEE providers. TEE providers could register/update their PPIDs in the TPM contract. Everyone could query the information (PPIDs and blockchain accounts) of registered TEE providers from the TPM contract.

```
1  contract TPM {
2      struct tee_info {
3          address tee_provider;
4          string ppid;
5      }
6      string contract_name;
7      tee_info[] tee_list;
8      constructor() {
9          contract_name = "TEE Provider Manager Contract";
10     }
11     // query TEE provider's address and PPID by its index
12     function get_tee_info(uint256 memory index) public
           view returns (address, string memory) {
13         return (tee_list[index].tee_provider, tee_list[
               index].ppid);
14     }
15     // TEE providers register their PPID
16     function register(string memory ppid) public {
17         tee_info tee = tee_info({tee_provider: msg.sender
               , ppid: ppid})
18         if (tee not in tee_list) {
19             tee_list.push(tee);
20         }
21     }
22     // TEE providers update their PPID
23     function update(uint256 memory index, string memory
            new_ppid) public {
24         tee_provider, _ = get_tee_info(index);
25         assert(tee_provider == msg.sender);
26         tee_list[index].ppid = new_ppid;
27     }
28 }
```

**Listing 1: TPM contract implementation code.**

## C SMARTCALL CONTRACT WITH MODEL INFERENCE

`TEE.inference()` is implemented by the precompiled contract technique and acts as a bridge that connects on-chain contract and off-chain model inference.

```
1  contract SmartCall {
2      address contract_owner;
3      address precompiled_contract_address = 0x6000;
4      InferencePrecompiled TEE;
5      // Instantiate a precompiled inference contract
6      constructor() {
7          contract_owner = msg.sender;
8          TEE = InferencePrecompiled(
               precompiled_contract_address);
9      }
10     // Call the precompiled inference function
11     function inference(string memory cmd) public view
            returns (string memory) {
12         return TEE.inference(cmd);
13     }
14 }
```

**Listing 2: `inference` function in `SmartCall` contract.**

## D DISCUSSION OF TEE SECURITY AND PERFORMANCE

Two main problems of TEEs, *i.e.*, security and performance, which could be the potential adoption and deployment challenges of SMART. Even though we do not aim to solve them in this paper, we can briefly discuss some existing solutions for addressing these issues.

**TEE side-channel attacks.** Many studies have revealed possible side-channel attacks in TEE. For example, SgxPectre [4] utilizes speculative execution side-channel vulnerabilities to compromise

the confidentiality of SGX enclaves. Cachezoom [34] introduces a cache side-channel attack that can track all memory accesses of SGX enclaves with high spatial and temporal precision. Wang *et al.* [49] also identify 8 potential memory side-channel attack vectors, ranging from TLB to DRAM modules. T-SGX [42] investigates controlled-channel attacks that malicious OS can track enclave accessed addresses by intentionally triggering page faults. Although many TEE side-channel attacks are discovered, there are many defenses to these attacks proposed at the same time [5–7, 42]. TEE is still a developing technology with security flaws, but it is undeniable that TEE is a promising and powerful hardware-assisted security guard. We believe that existing studies can improve the security design of TEE.

**TEE performance improvement.** Due to the limited memory size (about 96MB EPC memory size in SGXv1) and security check policies of SGX [29], model inference within SGX could cause frequent page swaps between secure and insecure memory, which largely increases the inference time overhead of some big AI models. There are mainly two lines of work focused on accelerating

AI inference in TEE. The first type of solution is to accelerate AI inference by designing dedicated computation/load/store strategies inside the TEE. For example, Occlumency [29] designs on-demand weights loading, memory-efficient inference and parallel processing pipelines to maximizing the AI inference performance running in SGX enclave. Another type of work is to accelerate AI inference by outsourcing partial workload. For example, Goten [36], Soter [41], and Slalom [46] propose secure GPU-outsourcing protocols to enable TEE communicating with GPU for speeding up AI training and inference. We believe these solutions can help the SMART framework increase the model inference speed inside TEE.

As the TEE develops, its performance becomes stronger and stronger. For example, SGXv2 can support up to 1TB of EPC memory, greatly reducing the overhead of paging [30]. Moreover, other TEE implementations, such as ARM Trustzone, have no limit on the memory size of the secure world [17], which is also conducive to enclave applications with large memory requirements. Therefore, the performance of TEE in the future is no longer an issue.

