# OpenReview forum: "Advancing Web 3.0: Making Smart Contracts Smarter on Blockchain"
_ACM.org/TheWebConf/2024/Conference — TheWebConf24_

### Official Review · Reviewer_HzVW · 2023-11-16

**Novelty:** 5
**Technical Quality:** 5

**Review:**

This article introduces an approach for integrating AI model inference with smart contract execution in Web3 applications. Model inference is computationally expensive in comparison to most smart contract executions and thus is impractical to be implemented directly in smart contract code. Furthermore, model inference can be non-deterministic which conflicts with the consensus mechanisms of most blockchain systems. The proposal here is to split deterministic smart contract code which can mutate state on chain from non-deterministic model inference, which is run on off-chain nodes that do not alter state or participate in consensus.

The technical quality of the system implementation, including the precompiled contracts and attestation service, is sound. My main question reading this paper is rather a more general issue of the utility of this approach for a decentralized system. I note a lack of a "decentralization" dimension in the star chart shown in Figure 2, but if it were there the SMART system would score low in contrast to an on-chain approach that validates the model inference output through consensus. My understanding of the system is that each time a contract is executed at step 4-a the TEE provider loads the model from off-chain storage. What happens if the storage service goes offline? At that point no TEE provider will be able to complete its execution.

Furthermore, there is no verification of the actual model output value--only validity of the signed quote. (step 8)  What if a compromised TEE provider simply gives incorrect model outputs? How would this system verify that? The security analysis only considers malicious TEE providers ability to affect system liveness. This once again highlights the lack of decentralization in this approach.

Fundamentally, given the reliance on available storage services to store the models, I wonder what makes the SMART system model better than making a Dapp that runs model inference on a private server, taking the output and sending that as input to a normal smart contract execution? In that case the blockchain node is not triggering the model inference, the client is but I am struggling to see how that materially changes the kinds of applications you can build. The benefit of an on-chain approach is that the inference is decentralized and the nodes arrive at consensus about the model output.

In terms of related work, approaches for on-chain inference are largely dismissed out of hand. However, quantized models can be used in resource-contrained embedded systems despite being somewhat degraded in accuracy. Are such methods being utilized, and what kinds of "smart" web3 applications might be enabled by smaller, on-chain models? It would be good to see a slightly more balanced take on this, as well as a discussion of limiations that come from your proposed method (putting it off-chain).

Overall, the paper is well written and I appreciate the sharing of source code and effort involved. I also appreciate the approach to implement in a plug-in manner in an existing EVM tech stack. The evaluation shows that off-chain computation is significantly more efficient, though this is not especially surprising. My main question/concerns stem from the issues stated above.

Additional questions:

Line 261: Ref to Ekiden says it is off-chain but Oasis Network (mainnet derived from Ekiden) uses TEEs for on-chain compute.

What is to stop the TEE provider from stealing the secret key of the private model (See sec 4.3, step 4-c)?

How do you meter gas for inference on TEE provider? Can repeated calls to TEE.inference() halt the chain?

"Note that the blockchain node could outsource multiple model inference tasks to TEE providers simultaneously, and wait for responses in an asynchronous mode, thus the off-chain inference does not congest block generation"  How is this possible since solidity is a synchronous language?

Because model inference is run in SGX does that mean it can only do CPU inference or is GPU inference possible?

Grammar issues:

Line 45: "are one of key" -> "are one of the key"
Line 82: "build more fancy Web 3.0 applications" is informal language
Line 144: "Even though current two most" -> "Even though the current two most"
Line 289: plural agreement
Line 497: "provide an quote" -> "provide a quote"
Line 1056: "to do evil" is informal language

### Update (10 Dec 2023)

I acknowledge that I have read the rebuttal.

**Questions:**

See questions in review above.

**Reviewer Confidence:**

3: The reviewer is confident but not certain that the evaluation is correct

**Scope:**

3: The work is somewhat relevant to the Web and to the track, and is of narrow interest to a sub-community

---

### Official Review · Reviewer_DehU · 2023-11-23

**Novelty:** 4
**Technical Quality:** 4

**Review:**

Rebuttal response
----------------------------
You have made an extensive effort to reply to my review, for which I am grateful. Unfortunately, it has only affected my view a little. Nevertheless: enough to (slightly) increase my grade on novelty.


Summary
----------------------------
The paper proposes a way to enable smart contracts to make use of AI output. They do so by proposing a new type of blockchain, one that can forward calls to a new party (TEE providers).



Weak points
----------------------------
- Scope

  This is neither a web paper, nor a security paper.

- Ethics

  This paper aims to find a way to enable the 2 currently most energy-sucking
  technologies work well together. It is highly likely that that is a net
  negative for society - a point that at least merits a discussion.

- Novelty

  There are already ways for smart contracts to interact with the world outside
  the blockchain. While these may be hacky, this paper makes no effort to
  distinguish its contribution from such efforts.



Overall evaluation
----------------------------
This is a weird paper. First of all: it has a scope problem. Its contribution is not a security contribution, but a programming paradigm contribution. Moreover, it has little to nothing to do with the Web. That implies that it is not that interesting for TheWebConference's security track.
Second of all: the introduction makes it sound as if including AI models in smart contracts is a desperately needed idea. But it fails to motivate that, from the smart contract end as well as from the AI end. Moreover, obviously AI models can much better be offered by cloud providers in an AI-as-a-service model. Of course, the paper recognises this and does the obvious thing: offload intense computation off the blockchain.
So, in effect, the paper proposes a way for on-chain code to interact with off-chain code. Nowhere near as lofty as is claimed.
Lastly: merging the two most energy-hogging technologies ever emerged from the field of CS raises moral questions -- questions that this paper fails to recognise.
All in all, I'm not a fan of the positioning of this paper. On top of that, I do not think it fits for WWW-SEC.



Comments for authors
----------------------------
- General:
  +   do not use "utilize". It leaves a rather bad impression.
  +   no spaces before footnotes.

- Intro:

  The idea to run AI on smart contracts seems patently stupid: this is not what
  blockchains / the EVM were designed for, nor what AI is designed for.
  Moreover, it is not clear what usecase this addresses: if one can interact
  with the blockchain, one could presumably interact with an AI-as-a-service
  provider. A solution desperately looking for a problem.

- pg 2, "TEE": acronym not explained. Just write it out full the first time.
- pg 2, footnote: this is not a grammatically well-formed sentence; its meaning
  is ambiguous.
- pg 2, "In specific,": this is not English. You probably meant "Specifically,"
- pg 3, Fig 2: on what do you base the scores for "on-chain" and "off-chain"?
  And why do you group all previous solutions into two diagrams?
  This figure comes across more like a poor attempt at hype than an actual
  comparison.
- pg 3, sec 2.2: note that Intel has deprecated SGX. Because it wasn't secure.
  That is: there is no ubiquitous secure enclave standard any more (well,
  turns out there never was). This takes most of the shine away from this
  section; you should at the very least acknowledge both SGX's deprecation and
  its inability to deliver on the assumptions you need for your implementation.
- Sec 3, "SMART"
  You should've called it "smartr contracts". Sounds better, is recognisable, and
  probably still trademarkable.
- Sec 3.2
  +   Methodology missing: how did you arrive at these threats?
  +   speaking of which: there are some threats missing.
      One of the missing ones concerns a TEE piece of code being malicious.

**Questions:**

-

**Ethics Review Description:**

No discussion of the detrimental societal impact of uniting two energy-hogging technologies.

**Ethics Review Flag:**

Yes

**Reviewer Confidence:**

2: The reviewer is willing to defend the evaluation, but it is likely that the reviewer did not understand parts of the paper

**Scope:**

1: The work is irrelevant to the Web

---

### Official Review · Reviewer_Sexm · 2023-11-26

**Novelty:** 6
**Technical Quality:** 4

**Review:**

Smart contracts are at the core of Web 3.0 and guaranteeing their correctness is paramount. In this work, the authors propose SMART, a smart contract framework that is compatible with current blockchains and at the same time supports efficient AI model inference. Given the complexity of existing AI models, the authors propose a solution combining on-chain execution for the deterministic part, and an off-chain execution to compute the complex AI models. The non-determinism derived from the AI models is dealt by leveraging TEEs in order to endorse the integrity and correctness of these off-chain executions, allowing this way all nodes to validate the offloaded computations. The authors also design distributed attestation and secret key provisioning schemes to allow the usage of private models. A prototype of SMART is presented and evaluated in EVM-based blockchains, and the authors show that it achieves not only the intended goals of correctness, liveness, and model privacy, but also that is more efficient that existing on-chain solutions.

The paper is well-written and well-structured providing a clear view of it goals. The theme is appropriate for the track, and timely with the widespread usage of LLMs.

Pros

- Proposal of a novel framework to support off-chain model inference
- Development of a prototype that is available online

Cons

- The paper would benefit from some clarifications as indicated in the comments
- The key-management committee is fixed making the system dependent on $t$ out of $n$ of these members

**Questions:**

- pg 5: V_TEE and H_enclave are used in section 4.1 but are not defined previously

- pg6: clarify what does it mean ``V_TEE satisfies the minimum value and H_enclave is the desired one."

- Section 4.3, secret key provisioning: It is not clear how the key-management committee is set. From the text there is a fixed set of $n$ TEE providers that form this committee, and that have shares for all the existing private models. However, this implies that no more than $n - t$ of these leave the network, otherwise it would be impossible to decrypt the private models

- As for the distribution of these keys, the distribution over trusted channels is unclear. Which secure channels exist between the client and the committee? Why are these communications trusted? It is not immediate why identity attestation between TEEs and clients guarantees the existence of secure channels

- A follow up, how are these keys loaded into the TEE?

**Reviewer Confidence:**

3: The reviewer is confident but not certain that the evaluation is correct

**Scope:**

3: The work is somewhat relevant to the Web and to the track, and is of narrow interest to a sub-community

---

### Decision · Program_Chairs · 2024-01-22

**Decision:**

Accept

**Comment:**

The paper proposes an architecture for blockchains where some of the off-chain computations are done by TEEs.
 Overall, the paper has a nice architecture and some interesting experimental results.
 From the novelt point of view such ideas have been proposed before and the suggested modifications seem rather incremental.

 ---